# Post-Traumatic Stress Disorder among Funeral Directors after the First Wave of COVID-19 in Belgium

**Roel Van Overmeire \*** , **Lara Vesentini** and **Johan Bilsen**

Mental Health and Wellbeing Research Group, Department of Public Health, Vrije Universiteit Brussel, 1090 Brussels, Belgium
\* Correspondence: roel.van.overmeire@vub.be

**Abstract:** (1) Background: Studies have shown that healthcare workers have an increased chance of developing post-traumatic stress symptoms (PTSS) during the COVID-19 pandemic. However, funeral directors have not yet been studied, despite facing many of the same possible causes of PTSS as healthcare workers. (2) Methods: An online survey was spread to directors of funeral home organization DELA in Belgium ($n = 287$). The validated scale PCL-5 was used to assess PTSS. Additionally, fear of being infected, fear of infecting family or friends and demographic factors (age, gender, living situations, work experience) were asked. Respondents were asked to indicate if they thought of the COVID-19 period when completing the PCL-5. In addition, with a self-made question, respondents were also asked to compare their current stress-levels with those before COVID-19. (3) Results: 107 respondents were included in the study. Thirteen respondents were above the threshold for possible PTSD-diagnosis, but only four fulfilled the criteria of completing the PCL-5 with COVID-19 in mind. PTSS-scores were significantly associated with a fear of being infected ($p = 0.006$). Almost a majority (49.5%) of the respondents felt more stress during the period of completing the survey than the period before COVID-19, though 33.6% felt no change. (4) Conclusions: Though the PTSS-symptom rates were lower than for healthcare workers, this study does show that there should be attention for the mental health of funeral directors in the aftermath of COVID-19.

**Keywords:** PTSD; funeral directors; COVID-19; mental health





## 1. Introduction

Post-traumatic stress disorder (PTSD) is a mental health disorder that, according to the *Diagnostic and Statistical Manual of Mental Disorders, fifth edition* (*DSM-V*), can be diagnosed after one month of specific symptoms after a traumatic event. These symptoms include avoidance-symptoms, flashbacks, feelings of detachment . . . [1]. In the context of COVID-19, there is disagreement on whether PTSD can be linked with having COVID-19. For example, the *DSM-V* does not recognize having a serious disease as a possible trauma, thus excluding having a disease as part of the inclusions for PTSD [2–5]. Thus, throughout, we will refer to PTSS: post-traumatic stress symptoms, as it remains unclear if the disease COVID-19 can be counted as part of the inclusion criteria for the diagnosis or screening of PTSD. It does however remain important to study such PTSS, as they can give indication of serious ailments, which, though not recognized as such by the *DSM-V*, should be treated. For example, PTSS have been found to be common among healthcare personnel during the COVID-19 pandemic, with 26.2% in Italy suffering from PTSS and in Norway even 28.9% [6–8]. The reasons why PTSS-rates might be so high among healthcare workers are numerous: work overload, exposure to the virus and fear of infecting family and friends. The highest PTSS-rates are generally among healthcare workers who care for patients [8,9].

However, a group that has been seldom studied is funeral directors. In general, there are few studies on funeral directors, with most dating from the 1990s or 2000s [10]. During

the pandemic, new studies have been performed, with studies in Canada, the United States and Belgium, with focusses on general wellbeing, compassion fatigue, anxiety and depression [11–14]. Yet, despite there being only a few studies having studied PTSS yet, there are indications that they might also be at risk of developing PTSS. Funeral directors received an overload of bodies during COVID-19, resulting in many even being kept in cooling trucks until they could have a proper funeral [10,11]. This enormous workload in combination with often a lack of proper protection equipment [15,16] could have led to funeral directors feeling threatened by serious illness or death. Furthermore, previous studies have shown that increased workloads might lead to more burnout or depressive symptoms among funeral directors [11–14]. Furthermore, a recent study with a small sample of ten funeral directors, showed that half of them possibly had PTSD during COVID-19 [17], while a master thesis showed that 28.5% of funeral directors in the U.S. met PTSD-criteria before COVID-19 [18]. The difficult and dangerous circumstances in which funeral directors work, and the mental health issues noticed in previous studies, indicate that PTSS might be a prevalent issue among funeral directors. Thus, it is important to study funeral directors, as they fulfill an important role in society: giving the proper funeral to those who died, despite the difficult work environment.

Thus, considering the indications that PTSS might be present among funeral directors, in this study we will investigate PTSS among funeral directors in Belgium during the first COVID-19 wave.

## 2. Methods

### 2.1. Design, Study Population and Data Collection

This study employs a cross-sectional design. An e-mail with a link to an online survey was sent to all 287 full-time funeral directors of DELA, the largest funeral home organization in Belgium (controls 11.5% of the funeral home industry in Belgium). They could complete the survey between 2 and 10 June 2020 and in three languages: English, Dutch and French. This was more than one month after the peak of 322 deaths on the 8th of April [19], which is important as PTSS can only be diagnosed one month of symptoms after a traumatic event (1). Between 2 and 10 June there were 120 deaths in total in Belgium, which was among the lowest of deaths since the beginning of the first wave of COVID-19 [19]. Through the use of internet cookies, the internet survey could not be filled in twice.

### 2.2. Measures

To measure PTSS, the PCL-5 (PTSD-checklist *DSM-5*) was employed, which is a validated scale, available in English, French and Dutch, and is based on the *DSM-V* criteria for PTSD. It contains 20 statements related to a traumatic event, and the sum of these statements leads to a range between 0 and 80 with the suggested cut-off for probable PTSS being between 31 and 33. In this study, 31 was chosen as the cut-off value, with scores above 31 being indicative of a probable PTSS, as a lower cut-off value is recommended to maximize detection of PTSS [20]. In the current study, the PCL-5 had a Cronbach alpha of 0.952, which was similar to a study on healthcare workers during COVID-19 ($\alpha = 0.94$) [6]. As the PCL-5 holds statements related to a traumatic event, respondents were asked if they had answered the PCL-5 statements with the first COVID-19 wave in mind.

In addition to demographic characteristics (age, gender, living situation) and work experience, respondents were asked to what extent they feared being infected themselves during their work and infecting their family members. These two questions were asked with a 5-point Likert scale, which was afterwards recoded to 'Low' and 'High', where 'High' is 4–5. Additionally, respondents were questioned on whether they had contact with the relatives of someone who died of COVID-19 and whether they were near the corpse of someone who died of COVID-19.

We also added a self-made question where we asked respondents to compare their current stress-levels with the period before COVID-19, where −5 equaled "Much more stress than the period before COVID", 0 meant "No stress changes" and +5 meant "Much

less stress than the period before COVID". In the results, we will present the results of this questions in the categories: $-5$ to $-1$ indicates more stress now than before, 0 indicates no changes in stress and $+1$ to $+5$ indicates less stress than before.

### 2.3. Analysis

As assumptions of normality were not met, non-parametric analyses were performed. Mann–Whitney U tests were used to compare categories based on the continuous PCL-5 score. This was not conducted for the question of comparing stress before and during COVID, as this question had three categories and would have required a Kruskal–Wallis test, for which our categories were too small in respondents.

If the PCL-5 was not answered in full, the respondent was left out of the analysis. Due to small counts, age was re-coded to the ages of 24–44 and 45–65. Living situation was recoded to 'people living with someone' (e.g., partner, children, parents) and 'people living alone'.

### 2.4. Ethics

The e-mail with the survey link was spread through DELA, but the link brought respondents to an anonymous online survey, inaccessible for DELA. Furthermore, the study was approved by the ethics commission of the VUB/UZ Brussels (B1432020000151).

## 3. Results

### 3.1. Characteristics of Respondents

Of 287 employees, 121 responded to the survey (42%). A total of 14 respondents of the 121 (11.8%) did not complete the PCL-5 scale and were left out of the analyses, leaving 107 respondents. Of these, 53.3% were female, 51.4% were older than 44 years old, 11.2% lived alone and 28% of the sample had 5 years or less work experience (Table 1).

**Table 1.** Demographic Characteristics and PCL scores.

|  | Total (N (%)) |
|---|---|
| Total | 107 (100) |
| **Gender** | |
| Male | 50 (46.7) |
| Female | 57 (53.3) |
| **Age** | |
| 24–44 | 52 (48.6) |
| 45–64 | 55 (51.4) |
| **Living situation** | |
| I live alone | 12 (11.2) |
| I live with someone | 95 (88.8) |
| **Work experience** | |
| <1–5 years | 30 (28) |
| 6–20 years | 77 (72) |
| **Contact or in presence of people who died due to COVID-19** | |
| No | 34 (31.8) |
| Yes | 73 (68.2) |
| **Contact with relatives of someone who of COVID-19** | |
| No | 22 (20.6) |
| Yes | 85 (79.4) |

**Table 1.** *Cont.*

| | Total (N (%)) |
|---|---|
| **Fear of getting infected** | |
| Low | 81 (75.7) |
| High | 26 (24.3) |
| **Fear of infecting family or friends** | |
| Low | 60 (56.1) |
| High | 47 (43.9) |
| **Stressful event in the PCL-5 related to COVID-19?** | |
| No | 64 (59.8) |
| Yes | 43 (40.2) |
| **Stress comparisons** | |
| More stress than before COVID-19 | 53 (49.5) |
| No changes in stress | 36 (33.6) |
| Less stress than before COVID-19 | 18 (16.9) |
| **PCL-5 score** | |
| < or =31 | 94 (87.9) |
| >31 | 13 (12.1) |
| PCL-mean score | 14.29 ($\pm$14.53) |

### 3.2. Exposure to COVID-19 and Fear of Infection

A total of 68.2% of the respondents were in the presence of someone who had died of COVID-19 and 79.4% had contact with relatives of someone who died of COVID-19. Only 24.3% of the sample feared getting infected with COVID-19 during work, whereas 43.9% feared infecting their family and/or friends.

### 3.3. Post-Traumatic Stress Symptoms

When completing the PCL-5, 40.2% of the sample considered the COVID-19 crisis as the reference point for their traumatic event. The average for the PCL-5 score was 14.29 ($\pm$14.53). There is a significant association between considering COVID-19 as a traumatic event and scoring higher on the PCL-5 scale ($p = 0.001$), meaning that someone who experienced COVID-19 as traumatic will also have more PTSS. However, of all 107 respondents, only 13 (12.1%) had scores above 31, meaning they screened positive for possible PTSS. Of these thirteen respondents with possible PTSS, only four viewed their work during the COVID-19 wave as their traumatic event to which they related the statement in the PCL-5 (3.3%).

### 3.4. Post-Traumatic Stress, According to Demographic Characteristics, Work Experience, Exposure, Fear of Infection

Using Mann–Whitney U tests, PCL-5 scores were compared for categories. No significant associations were found for gender ($p = 0.420$), work experience ($p = 0.274$), age ($p = 0.356$) or living situation ($p = 0.07$). Being in contact or in the presence of people who have died of COVID-19 ($p = 0.965$) or with their family members ($p = 0.258$) was also not significantly associated with PCL-5 scores (Table 2).

Respondents who were afraid of being infected had significantly higher PCL-5 scores than people who were not afraid ($p = 0.006$). There was no significant association concerning the fear of infecting family or friends ($p = 0.062$).

**Table 2.** Associations between PCL-5 and Categories.

| | PCL-5 Score < or = 31 (N (%)) | PCL-5 Score > 31 (N (%)) | PCL-5 (Mean Rank) | *p*-Value * |
|---|---|---|---|---|
| Total | 94 (100) | 13 (100) | | |
| Gender | | | | |
| Male | 42 (44.7) | 8 (61.5) | 51.42 | 0.420 |
| Female | 52 (55.3) | 5 (38.5) | 56.26 | |
| Work experience | | | | |
| <1–5 years work experience | 45 (47.9) | 7 (53.8) | 48.75 | 0.274 |
| 5–20 years work experience | 49 (52.1) | 6 (46.2) | 56.02 | |
| Age | | | | |
| 24–44 years | 10 (10.6) | 2 (15.4) | 56.85 | 0.356 |
| 45–64 years | 84 (89.4) | 11 (84.6) | 51.31 | |
| Living situation | | | | |
| Living alone | 29 (30.9) | 1 (7.7) | 69.29 | 0.07 |
| Living with someone | 65 (69.1) | 12 (92.3) | 52.07 | |
| Contact or in presence of people who died due to COVID-19 | | | | |
| No | 30 (31.9) | 4 (30.8) | 53.81 | 0.965 |
| Yes | 64 (68.1) | 9 (69.2) | 54.09 | |
| Contact with relatives of someone who of COVID-19 | | | | |
| No | 18 (19.1) | 4 (30.8) | 47.34 | 0.258 |
| Yes | 74 (80.9) | 9 (69.2) | 55.72 | |
| Fear of getting infected | | | | |
| Low | 74 (78.7) | 7 (53.8) | 49.36 | 0.006 |
| High | 20 (21.3) | 6 (46.2) | 68.46 | |
| Fear of infecting family or friends | | | | |
| Low | 54 (57.4) | 6 (46.2) | 49.06 | 0.062 |
| High | 40 (42.6) | 7 (53.8) | 60.31 | |
| Stressful event in the PCL-5 related to COVID-19? | | | | |
| No | 55 (58.5) | 9 (69.2) | 45.48 | 0.001 |
| Yes | 39 (41.5) | 4 (30.8) | 66.67 | |

* *p*-value based on Mann–Whitney U tests.

### 3.5. Comparisons of Stress

In total, 49.5% had more stress during COVID-19 than before, 33.6% experienced no changes and 16.9% experienced less stress during COVID-19 than before.

## 4. Discussion

In the current study, PTSS was investigated among funeral directors in Belgium, more than one month after the peak of infections and deaths due to COVID-19. Of the sample, 12.3% (*n* = 13) had scores indicative of possible PTSS, though only four respondents also indicated that they viewed COVID-19 as a possible traumatic event. Indicative of a high PTSS score was the fear of getting infected, though not the fear of infecting family or friends. Most respondents (49.5%) also experienced more stress during COVID-19 than before.

The percentage of funeral directors with possible PTSS (12.3%) was not high when compared to studies on healthcare workers during COVID-19. One study in Italy found 26.2% of a sample suffering from PTSS [8], and a study in Norway found 28.9% [6]. Then again, in a pre-corona study in the U.S. in a representative sample, a prevalence of only 3.4% was found in the general population [21]. So, it might seem that funeral directors do have high rates, but then again, the study by Cloitre et al. [21] used the *ICD-11* definition, which has been known to give different results than the *DSM-V* definition [22]. Furthermore, that

study also used inclusion criteria for PTSD (e.g., experiencing traumatic events). If we apply inclusions, only four respondents fulfilled PTSS-criteria, meaning only 3.3%. If we compare it to the one COVID-19 related study on PTSD among funeral directors during COVID-19, we see that 5 out of the 10 respondents (50%) had possible PTSD. While it is difficult to make any general assumptions based on this result, it is of course clear that our results are far below the 50% found by Hicks et al. [17]. When compared to another study in the U.S. shortly before COVID-19, our results are still far below the PTSD-rates found (namely, 12.3% to 28.5%) [18]. Thus, from whatever way we view these results, PTSS were quite low among funeral directors during the first wave of COVID-19.

How can these results be explained? First, there is the national context. The study by Durand-Moreau and Galarneau [14] showed high rates of anxiety and depression among Canadian funeral directors. Similarly, Cegelka et al. [23] showed that American funeral directors also suffer from wellbeing issues. Such results might indicate that the Belgian context was simply less extremely stressful for funeral directors. Second, which is much more difficult to assess, is the internal management styles of different funeral organizations. We studied DELA, a large organization, which is considered by some company-studies to be among the best employers in Belgium, because it supposedly is a good environment to work in and offers psychosocial care to their employees [24]. Thus, it might actually also have led to lower rates, not because funeral directors in Belgium do not have stressful work, but because the work environment provides a buffer against developing severe problems. That would explain why almost the majority experienced more stress during COVID-19 (which is a normal reaction), but only 12.3% developed PTSS. It might be that funeral organizations in other countries cannot offer the same environment to their employees. For example, in the U.S. study by Hicks et al. [17], 30% had no or little knowledge of mental health resources for funeral directors—though this was a study with only ten respondents. It is also unknown if employees have used the psychosocial care offered by DELA and in what way all the funeral directors would agree with DELA being a good employer. Third, there was quite a large geographical spread of COVID-19 victims in Belgium, meaning that some regions had higher COVID-19 casualties than other regions, which might have resulted in generally lower work pressure and, thus, perhaps fewer stress-related problems. Fourth, PTSD-scales generally have a higher cut-off than other scales. For example, depression and anxiety scales have a generally low cut-off point. For example, the PHQ-2, which is a quick screening for depression, is 2 questions, with a score from zero to six, where two and above is indicative of possible depression [25], whereas the PCL-5 has 20 questions and perhaps a higher cut-off point. That the threshold to be screened positive for PTSS/PTSD might be higher than for other disorders can also be seen in that while almost half the sample experienced more stress, only 12.3% actually screened positive for PTSS. However, while these observations might explain the differences between the different funeral director studies, it does not explain the differences in PTSS rates between healthcare studies and the current studies, which brings us to our final point. Since this is one of the first studies on funeral directors and PTSS, there is no comparison possible between other studies, and thus we have made comparisons with healthcare workers. However, it might be that healthcare workers simply had a lot more severely stressful situations during COVID-19 than funeral directors. This is not an unreasonable assumption to make, considering that while there will have been undoubtedly a work overload for some funeral directors, the issues that healthcare workers faced were not just work overload, but also regularly seeing patients dying. The experience of seeing a patient die might have been simply much more direct and stressful than the process of treating some deceased. This explanation does not exclude that funeral directors can experience mental health issues during COVID-19—in fact, multiple studies have shown as much (e.g., [14,23]). It does, however, indicate that perhaps the extreme stress reactions associated with PTSD are not present among funeral directors. In addition, not every respondent in our sample considered COVID-19 as their reference period for the completion of the PTSD-scale. Thus, not all funeral directors necessarily associate the period of COVID-19 with an incredibly stressful period. In fact, 33.6% even felt

no difference in work stress compared to the period before COVID-19. Furthermore, only 24.3% actually feared infection from COVID-19. Naturally, mental health issues can arise from other problems. In fact, population studies show that mental health issues also rose in the general population, indicating that the restrictions of COVID-19 could cause issues such as depression [26,27], which therefore might also have their effect on funeral directors. Furthermore, the job satisfaction of funeral directors might also have suffered from the restrictions that were imposed on their work. For example, funeral services were cut short and only immediate family was allowed to be present [10]. Thus, such circumstances might better explain the relatively low PTSS-rates, though other studies have indicated more depression and anxiety among funeral directors. However, this does not mean that there should be no attention for the mental health of funeral directors. Again, several studies have shown that there is reason for concern. Furthermore, PTSS among 12.1% is still a lot; it is simply a lot lower than among healthcare workers. To end this part of the discussion, we might conclude that in interpreting PTSS-results we must always look at, of course, the methodology (e.g., How is PTSS measured? Who is included?), but equally important, the context in which the PTSS-results are found. In this study, for example, multiple contextual factors might have played a role, such as the geographical spread, the internal management of the organization, the national context . . .

One of the most important results in this study is the significant link between PTSS-rates and the fear of being infected with COVID-19. It might be that there is an objectively higher chance of funeral directors getting infected, as not all funeral directors receive the proper training to handle bodies and do not always know of what someone died (e.g., what disease), increasing the chances of infection [15,16]. Though, an important side note to this is that only 24.3% feared getting infected themselves. It would indicate that funeral directors generally were feeling quite safe, but that it is among the 24.3% that fear infection that PTSS is most associated. Furthermore, while 43.9% feared infecting their family and friends, indicating that they feared that friends and family might indeed have a lack of proper protection, there was no clear association with PTSS.

This study was one of the first to assess PTSS among funeral directors, and the first to find an association between fear of getting infected with COVID-19 and PTSS among funeral directors. It is also one of the few studies on funeral directors during COVID-19 and funeral directors in general [10].

However, there are several limitations to this study. First, the sample size limited our analysis possibilities, and the possibility to generalize it to the entire population of funeral directors. Especially with regard to the divide of non-PTSS and probable PTSS we were hindered by the fact that there were not many who had probable PTSS. Second, other factors that might impact the mental health of funeral directors, but that were not investigated in this study, are the fear of death, occupational death exposure, the number of funerals and stressful incidents [28–31]. It is quite possible that these factors are present more than ever during this COVID-19 crisis. Furthermore, considering the nature of the questionnaire, it might be that people who were suffering from PTSS might have chosen to avoid the questionnaire. Third, the comparison of stress during and before COVID-19 was hampered by not being validated and of course being quite biased in the assessment. It is after all difficult to estimate if someone has more stress now than before COVID-19. It also said little about the stress someone experiences at that moment, only if it was more or less than before in their estimation. Due to the limitations of this question, and because our focus was on PTSS, we have avoided discussing it as much as the PTSS results. Finally, the biggest limitation in terms of screening and diagnostics, is that there is uncertainty of the diagnosis of PTSS in relation to COVID-19. In the *DSM-V*, PTSD is not supported as a diagnosis under the circumstances of a pandemic [2,32]. In fact, the argument could be made that what we should have been researching was not PTSS but adjustment disorders [32]. Adjustment disorders are quite similar to PTSD but are related to non-traumatic stressful events. In fact, the *ICD-11* definition of adjustment disorders even clearly indicated why it might be more associated with COVID-19 circumstances: "*Adjustment disorder is a maladaptive*

*reaction to an identifiable psychosocial stressor or multiple stressors (e.g., divorce, illness or disability, socio-economic problems, conflicts at home or work) that usually emerges within a month of the stressor*" [33]. Illness, socio-economic problems and conflicts at work are situations that we might associate with COVID-19 circumstances. Furthermore, adjustment disorder research is quite underrepresented in the scientific literature, despite being diagnosed regularly by psychiatrists [34]. Yet, the reason why it was not investigated in our study is the current limitations of adjustment disorder scales: there are only a few presently tested and often they have not been applied enough in studies to allow comparisons. For example, it would have been impossible to compare the results of the current study to any other study, because there are relatively few studies on adjustment disorders.

Managers of funeral homes should be aware of the possible danger of funeral directors having mental health problems or even PTSS, although this latter risk is rather low. There are numerous ways in which funeral directors can be prepared to deal with extreme stress during COVID-19. First, the social support of peers is important. Second, it is not clear how the fear to get infected can be countered, as it seems difficult to eliminate the fear to get infected when media constantly report on infections. However, management in funeral homes can try to counter it by training on protective measures (e.g., how to handle the body of someone that died of COVID-19) to give them a sense of safety.

Further studies should concentrate on the events that influence the mental health of funeral directors, and more precisely, the fear of getting infected. Perhaps there is a relationship with the perception of protective measures: if protective measures are seen as insufficient, there might be a higher fear to get infected. Furthermore, the fear to get infected might be interesting to explore in other studies.

## 5. Conclusions

An important result in this study was that while more funeral directors feared infecting their family and/or friends with COVID-19, it was the association between being infected themselves and development of PTSS that was significant. However, when compared with healthcare workers, PTSS rates were quite low among funeral directors. While almost a majority of the funeral directors experienced more stress than before COVID-19, this did not translate into high PTSS rates. This indicated that in general, funeral directors in Belgium might be less vulnerable for developing PTSS, perhaps due to the nature of their job or better psychosocial care.

**Author Contributions:** Conceptualization, R.V.O.; methodology, R.V.O.; formal analysis, R.V.O. and L.V.; data curation, R.V.O. and L.V.; writing—original draft preparation, R.V.O.; writing—review and editing, R.V.O., L.V. and J.B.; supervision, J.B. All authors have read and agreed to the published version of the manuscript.

**Funding:** This research received no external funding.

**Institutional Review Board Statement:** The e-mail with the survey link was spread through DELA, but the link brought respondents to an anonymous online survey, inaccessible for DELA. Furthermore, the study was approved by the ethics commission of the VUB/UZ Brussels (B1432020000151).

**Informed Consent Statement:** In the introduction screen of the online survey, participants were informed about their rights with regard to the survey (e.g., no obligation to complete the survey) and that DELA, their employer, would have no insight into their answers. Furthermore, they were informed that if they would start the survey, they would be giving an implied consent and that all answers are completely anonymous.

**Data Availability Statement:** Data is contained within the article.

**Acknowledgments:** We wish to thank the employees of DELA for their effort in completing this survey.

**Conflicts of Interest:** The authors declare no conflict of interest.

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
