# Peer review of "Post-Traumatic Stress Disorder among Funeral Directors after the First Wave of COVID-19 in Belgium"

_psych, doi:10.3390/psych4030046_

Round 1

Reviewer 1 Report

This article is very interesting. However, it is important to compare two groups doing work at a similar level of commitment, that is, the work of health care workers to funeral home workers. Comparing directors , who are less involved in contact with COVID-19 to health care workers, who have direct contact with the sick, will not allow to draw such conclusions. It needs to be determined precisely whether the study groups had regular contact with patients infected with or who died from COVID-19.

Should be completed author contribution

the bibliography should be written correctly

Author Response

This article is very interesting. However, it is important to compare two groups doing work at a similar level of commitment, that is, the work of health care workers to funeral home workers. Comparing directors , who are less involved in contact with COVID-19 to health care workers, who have direct contact with the sick, will not allow to draw such conclusions. It needs to be determined precisely whether the study groups had regular contact with patients infected with or who died from COVID-19.

ANSWER: We thank the reviewer for the kind words. We agree, and indeed, neither in terms of PTSD, nor in terms of contact with people who died from COVID-19 are healthcare workers and funeral directors really comparable. We have chosen healthcare workers as main reference point because there is simply no possibility in comparing with other funeral director-studies in terms of PTSD-rates, as there is only one other study, with a sample of only ten respondents. We also try to highlight why healthcare workers are obviously more inclined to develop PTSD.  

We also have the data on their contact with people who died from COVID-19. It is impossible to determine whether or not they had regular contact or not. But, we do believe our data give a clear overview of the general contact between funeral directors and people who died from COVID-19 or even family members of people who died from COVID-19, as well as their own fears of getting COVID-19. 

Should be completed author contribution

ANSWER: This has been done.

the bibliography should be written correctly

ANSWER: This has been adjusted.

Reviewer 2 Report

Overall Impressions: This is an interesting study that is severely limited by the fact that it is just looking at Belgian funeral home directors and the association between the COVID-19 period and PTSS symptoms measured using the validated PCL-5 measure. If the study had been carried out in the United States and Italy and other place much more hard hit by the pandemic, this reviewer believes much higher rates of PTSS or PTSD would have been observed, as has been seen in other studies this reviewer found carried out in the US. The really surprising thing in this study is that the rates of PTSS or PTSD (PCL-5 can also used to measure PTSD) are so low in Belgian funeral home directors. In the few other studies that exist, much higher rates of actual PTSD were observed. Although the study is of limited value because of the the choice to only sample Belgian funeral home directors, it is a decent sample size and perhaps may help to inform much larger future international studies of funeral home workers in general, who already have much higher levels of PTSD compared to the general population as measured for example in the Unites States. The study would have increased value if there were a comparison of PTSD or PTSS in funeral home directors in Belgium compared to the general population or to health care workers in Belgium in order to provide some sort of baseline for comparison with the general population, as it has been shown in the US that mortuary and funeral home workers have a rate of PTSD three times higher than the general population (https://scholarworks.umt.edu/gsrc/2022/326/7/).

Even given all the above critiques, the authors clearly have a strong understanding of PTSS and PTSD criteria and evaluation and have laid out some important arguments for why there was so little PTSS/PTSD found in funeral home directors in Belgium, which might be of value on an international scale, such as the DELA organization making sure that funeral home directors and employees are well taken care of and have substantial access to mental health care and other types of support services, such as support groups, time off, increased hazard pay, etc. Also, the discussion concerning higher rates of PTSS/PTSD in healthcare workers in Belgium versus funeral directors in Belgium is also well thought out and well argued, and could lead to further studies.

Line 31-33: Abstract: There seems to be one fundamental limitation this study which is rather than delivering the survey at the actual point in the pandemic, the authors ask that the surveyants “think” about the COVID-19 pandemic at that point after the fact; this seems like a major structural problem that could change the results of the valildated survey. Rather than being able to deliver the PCL-5 survey at the time of the COVID-19 period of interest would weaken the strength of the findings, most likely underestimating the scores that would have been registered at the time.

“Respondents were asked to indicate if they thought of the COVID-19 period when completing the PCL-5. In addition, respondents were also asked to compare their current stress-levels with those before COVID-19.”

Line 35-37: It is unclear what is meant by this statement:

“Thus, throughout, we will refer to PTSS: post-traumatic stress symptoms, as it remains unclear if the disease COVID-19 can be counted as part of the inclusion criteria for the diagnosis or screening of PTSD.”

It is the understanding of this reviewer that it was the environment surrounding the pandemic of COVID-19 in many countries (especially Italy and the United States, for example) that created the possibility for PTSD, not whether you had an infection if COVID-19, and the hugely stressful environment that was created because of the COVID-19 pandemic had major effects on mental health, including anxiety, depression, PTSD, suicidality, during the beginning and well into the pandemic and beyond, and this has been well documented by the World Health Organization and many other large academic health centers and government bodies around the world. It is not clear what point the authors tare trying to make here about inclusion criteria for PTSD.  It is clear that the COVID-19 pandemic absolutely qualifies as a traumatic event, similar to exposure to war zones, natural catastrophes, major disease outbreaks, etc.

Line 46: It appears that the following statement may not be correct and should be amended: “However, a group among whom it has not yet been studied, are funeral directors.” A simple web search revealed that indeed PTSD had been studies in funeral home workers/mortuary workers (it is also unclear why the authors are so focused on Funeral Home Directors as opposed to Funeral Home Workers and Employees who actually perform most of the tasks carried out in funeral homes), and in a Masters Thesis at Harvard University 28.5% of 333 mortuary/funeral home workers met the criteria for PTSD (https://blog.frontrunnerpro.com/mental-health-funeral-industry/) and (https://dash.harvard.edu/bitstream/handle/1/37365094/MCCLANAHAN-DOCUMENT-2019.pdf?sequence=1)

When we think of PTSD, many people instantly associate this with military or first . While it is prevalent in those professions, PTSD among funeral professionals often gets overlooked and undiagnosed.

To date, there hasn’t been much research on the topic in regards to funeral professionals. With that being said, I was able to come across one intriguing research article online. In 2019, Jessica McClanahan, a student at Harvard University published her Master’s thesis examining post-traumatic stress among mortuary works. Click here if you want to read her full thesis.

For her research project, McClanahan interviewed 333 mortuary workers who identified that they were actively working in the profession. The results showed that 28.5% of the respondents met the criteria for PTSD. Comparatively, other studies on the general population have concluded that PTSD rates range from 1% (Helzer et al. 1987) to 8.3% (Kilpatrick et al. 2013). Regardless, this study suggests that PTSD rates among funeral professionals could be 20% or higher than the general population.”

McClanahan, Jessica. 2019. Posttraumatic Stress Among Mortuary Workers: Prevalence, Risk,

and Resilience. Master's thesis, Harvard Extension School.

Line 46-48: It is completely unclear why these authors are so focused just on “Funeral Home Directors” versus “Funeral Home and Mortuary Workers” who perform most of the actual work at these organizations? Also the self-citation of the last work published on Funeral Home Directors again begs the question of why these individuals are so focused on funeral home directors and not individuals who work at Funeral Homes and Mortuaries? “However, a group among whom it has not yet been studied, are funeral directors. In general, there are few studies on funeral directors, with most dating from the 90s or 00s 47 (Van Overmeire & Bilsen, 2020).”

Line 46-64: It appears that the authors have chosen to do a great deal of self-citation, which is actually a criteria to be examined in any research manuscript review, as it is not viewed positively. It may be the case that these authors are self-citing because they are only one of a very few groups that are studying the mental health of funeral home directors, and in this case this reviewer is more understanding of this practice, although it is generally frowned upon.

Line 55-56: Replace “This enormous workload in combination with often a lack of proper protection material” with “This enormous workload in combination with often a lack of proper personal protective equipment.”

Line 60-62: “Yet, despite the difficult and dangerous circumstances in which funeral directors work, and despite the mental health issues noticed in previous studies, it remains unclear if PTSS are present.” In fact, PTSD has been documented in funeral home directors as mentioned in the above Harvard Masters Thesis published study and in other smaller studies, for example, where the same PCL-5 measure was used:

Hicks, E., Mashinchi, G., Heppner, H., Bean, P., & Maxson, C. (2022). PTSD Prevalence Rates of US Funeral Directors during the COVID-19 Pandemic: A Brief Report. Advanced Journal of Social Science, 10(1), 45–51. https://doi.org/10.21467/ajss.10.1.45-51

This study also found that there were higher rates of PTSD among funeral home and mortuary workers even before COVID-19:

“Therefore, it is likely that funeral directors are also experiencing increased mental health concerns. Additionally, a recent study found that 28.5% of a sample of funeral directors studied met criteria for PTSD (McClanahan, 2019), a rate approximately 20% higher than that of the general population (National Center for PTSD, n.d.). Given the findings of high rates of PTSD for mortuary workers during typical times, and heightened job demands and burnout during the COVID-19 crisis, the present study sought to investigate PTSD rates of funeral directors during the ongoing pandemic. To our knowledge, it is the first study of its kind.”

Line 152: Note that in Table 2 for the first 2 PCL-5 Score columns in the table, all of the numbers are out of register with the associated description on the left hand side of the table. Please make sure the description and the actual numbers line up horizontally with each other.

Line 250-252: “This study was to our knowledge the first to assess PTSS among funeral directors, 250 and the first to find an association between fear of getting infected with COVID-19 and 251 PTSS among funeral directors.” It has been documented using the same PCL-5 scale in other studies that mortuary and funeral home workers and directors experience PTSD. Suggesting that this is the first study to look at PTSS does not seem to have any higher impact or meaning than the studies that have looked at PTSD directly.

Line 307-371: Why have the authors left the “Instructions to authors in many of these unfinished sections? These sections should have been finished and the instructions to authors removed before being sent our for review.

Author Response

REVIEWER 2:

Overall Impressions: This is an interesting study that is severely limited by the fact that it is just looking at Belgian funeral home directors and the association between the COVID-19 period and PTSS symptoms measured using the validated PCL-5 measure. If the study had been carried out in the United States and Italy and other place much more hard hit by the pandemic, this reviewer believes much higher rates of PTSS or PTSD would have been observed, as has been seen in other studies this reviewer found carried out in the US. The really surprising thing in this study is that the rates of PTSS or PTSD (PCL-5 can also used to measure PTSD) are so low in Belgian funeral home directors. In the few other studies that exist, much higher rates of actual PTSD were observed. Although the study is of limited value because of the the choice to only sample Belgian funeral home directors, it is a decent sample size and perhaps may help to inform much larger future international studies of funeral home workers in general, who already have much higher levels of PTSD compared to the general population as measured for example in the Unites States. The study would have increased value if there were a comparison of PTSD or PTSS in funeral home directors in Belgium compared to the general population or to health care workers in Belgium in order to provide some sort of baseline for comparison with the general population, as it has been shown in the US that mortuary and funeral home workers have a rate of PTSD three times higher than the general population (https://scholarworks.umt.edu/gsrc/2022/326/7/).

ANSWER: Indeed, the results might be different if the study was performed in the U.S. or Italy. However, for further studies it might interesting then to compare studies in those countries with our results. This is of course a limitation we cannot amend.

Even given all the above critiques, the authors clearly have a strong understanding of PTSS and PTSD criteria and evaluation and have laid out some important arguments for why there was so little PTSS/PTSD found in funeral home directors in Belgium, which might be of value on an international scale, such as the DELA organization making sure that funeral home directors and employees are well taken care of and have substantial access to mental health care and other types of support services, such as support groups, time off, increased hazard pay, etc. Also, the discussion concerning higher rates of PTSS/PTSD in healthcare workers in Belgium versus funeral directors in Belgium is also well thought out and well argued, and could lead to further studies.

ANSWER: We thank the reviewer for the kind remarks.  

Line 31-33: Abstract: There seems to be one fundamental limitation this study which is rather than delivering the survey at the actual point in the pandemic, the authors ask that the surveyants “think” about the COVID-19 pandemic at that point after the fact; this seems like a major structural problem that could change the results of the valildated survey. Rather than being able to deliver the PCL-5 survey at the time of the COVID-19 period of interest would weaken the strength of the findings, most likely underestimating the scores that would have been registered at the time.

ANSWER: We did so because PTSD criteria in DSM-V stipulate a one month duration of symptoms. The heigh of deaths in COVID-19 happened around a month before. So, the period was really chosen with a specific reason. But we understand the comment of the reviewer in this regard, and indeed, there might be a form of recall bias in this sense.

“Respondents were asked to indicate if they thought of the COVID-19 period when completing the PCL-5. In addition, respondents were also asked to compare their current stress-levels with those before COVID-19.”

Line 35-37: It is unclear what is meant by this statement:

ANSWER: The statement is somewhat confusing because of the two unrelated sentences following each other (because it is quite difficult to differentiate in an abstract). This refers to the self-made question on stress. We have changed it to “In addition, with a self-made question, respondents were also asked to compare their current stress-levels with those before COVID-19.”.

“Thus, throughout, we will refer to PTSS: post-traumatic stress symptoms, as it remains unclear if the disease COVID-19 can be counted as part of the inclusion criteria for the diagnosis or screening of PTSD.”

It is the understanding of this reviewer that it was the environment surrounding the pandemic of COVID-19 in many countries (especially Italy and the United States, for example) that created the possibility for PTSD, not whether you had an infection if COVID-19, and the hugely stressful environment that was created because of the COVID-19 pandemic had major effects on mental health, including anxiety, depression, PTSD, suicidality, during the beginning and well into the pandemic and beyond, and this has been well documented by the World Health Organization and many other large academic health centers and government bodies around the world. It is not clear what point the authors tare trying to make here about inclusion criteria for PTSD.  It is clear that the COVID-19 pandemic absolutely qualifies as a traumatic event, similar to exposure to war zones, natural catastrophes, major disease outbreaks, etc.

ANSWER: As there is currently still a lot of discussion on how COVID-19 fits into criterion A of the DSM-V, we have avoided calling it PTSD and instead PTSS. While seemingly semantic, we do believe this is diagnostically speaking the safer option.

Line 46: It appears that the following statement may not be correct and should be amended: “However, a group among whom it has not yet been studied, are funeral directors.” A simple web search revealed that indeed PTSD had been studies in funeral home workers/mortuary workers (it is also unclear why the authors are so focused on Funeral Home Directors as opposed to Funeral Home Workers and Employees who actually perform most of the tasks carried out in funeral homes), and in a Masters Thesis at Harvard University 28.5% of 333 mortuary/funeral home workers met the criteria for PTSD (https://blog.frontrunnerpro.com/mental-health-funeral-industry/) and (https://dash.harvard.edu/bitstream/handle/1/37365094/MCCLANAHAN-DOCUMENT-2019.pdf?sequence=1)

“When we think of PTSD, many people instantly associate this with military or first . While it is prevalent in those professions, PTSD among funeral professionals often gets overlooked and undiagnosed.

To date, there hasn’t been much research on the topic in regards to funeral professionals. With that being said, I was able to come across one intriguing research article online. In 2019, Jessica McClanahan, a student at Harvard University published her Master’s thesis examining post-traumatic stress among mortuary works. Click here if you want to read her full thesis.

For her research project, McClanahan interviewed 333 mortuary workers who identified that they were actively working in the profession. The results showed that 28.5% of the respondents met the criteria for PTSD. Comparatively, other studies on the general population have concluded that PTSD rates range from 1% (Helzer et al. 1987) to 8.3% (Kilpatrick et al. 2013). Regardless, this study suggests that PTSD rates among funeral professionals could be 20% or higher than the general population.”

McClanahan, Jessica. 2019. Posttraumatic Stress Among Mortuary Workers: Prevalence, Risk,

and Resilience. Master's thesis, Harvard Extension School.

ANSWER: We have added the master thesis to the manuscript and thank the reviewer for the extra references.

Line 46-48: It is completely unclear why these authors are so focused just on “Funeral Home Directors” versus “Funeral Home and Mortuary Workers” who perform most of the actual work at these organizations? Also the self-citation of the last work published on Funeral Home Directors again begs the question of why these individuals are so focused on funeral home directors and not individuals who work at Funeral Homes and Mortuaries? “However, a group among whom it has not yet been studied, are funeral directors. In general, there are few studies on funeral directors, with most dating from the 90s or 00s 47 (Van Overmeire & Bilsen, 2020).”

ANSWER: This mainly follows from our earlier viewpoint on the importance of studying such groups. Thus, in that sense our current study is simply the extension of the goal stipulated then.

Line 46-64: It appears that the authors have chosen to do a great deal of self-citation, which is actually a criteria to be examined in any research manuscript review, as it is not viewed positively. It may be the case that these authors are self-citing because they are only one of a very few groups that are studying the mental health of funeral home directors, and in this case this reviewer is more understanding of this practice, although it is generally frowned upon.

ANSWER: We understand this point completely, and we would not do so if it was possible to reference other studies. However, there is not a lot of literature on funeral directors’ mental health world-wide.

Line 55-56: Replace “This enormous workload in combination with often a lack of proper protection material” with “This enormous workload in combination with often a lack of proper personal protective equipment.”

ANSWER: This is done.  

Line 60-62: “Yet, despite the difficult and dangerous circumstances in which funeral directors work, and despite the mental health issues noticed in previous studies, it remains unclear if PTSS are present.” In fact, PTSD has been documented in funeral home directors as mentioned in the above Harvard Masters Thesis published study and in other smaller studies, for example, where the same PCL-5 measure was used:

Hicks, E., Mashinchi, G., Heppner, H., Bean, P., & Maxson, C. (2022). PTSD Prevalence Rates of US Funeral Directors during the COVID-19 Pandemic: A Brief Report. Advanced Journal of Social Science10(1), 45–51. https://doi.org/10.21467/ajss.10.1.45-51

ANSWER: We have added both Hicks, et al. and the master thesis to the manuscript. Again, thank you for the extra references. We think the Hicks-article might have appeared around the time of finishing our manuscript, which would explain why we had not been able to add it yet.

This study also found that there were higher rates of PTSD among funeral home and mortuary workers even before COVID-19:

“Therefore, it is likely that funeral directors are also experiencing increased mental health concerns. Additionally, a recent study found that 28.5% of a sample of funeral directors studied met criteria for PTSD (McClanahan, 2019), a rate approximately 20% higher than that of the general population (National Center for PTSD, n.d.). Given the findings of high rates of PTSD for mortuary workers during typical times, and heightened job demands and burnout during the COVID-19 crisis, the present study sought to investigate PTSD rates of funeral directors during the ongoing pandemic. To our knowledge, it is the first study of its kind.”

ANSWER: We have added the source of McClanahan, 2019.

Line 152: Note that in Table 2 for the first 2 PCL-5 Score columns in the table, all of the numbers are out of register with the associated description on the left hand side of the table. Please make sure the description and the actual numbers line up horizontally with each other.

ANSWER: This has been corrected.

Line 250-252: “This study was to our knowledge the first to assess PTSS among funeral directors, 250 and the first to find an association between fear of getting infected with COVID-19 and 251 PTSS among funeral directors.” It has been documented using the same PCL-5 scale in other studies that mortuary and funeral home workers and directors experience PTSD. Suggesting that this is the first study to look at PTSS does not seem to have any higher impact or meaning than the studies that have looked at PTSD directly.

ANSWER: We have adjusted it to: “This study was one of the first to assess PTSS among funeral directors, and the first to find an association between fear of getting infected with COVID-19 and PTSS among fu-neral directors. It is also one of the few studies on funeral directors during COVID-19 and funeral directors in general.”

Line 307-371: Why have the authors left the “Instructions to authors in many of these unfinished sections? These sections should have been finished and the instructions to authors removed before being sent our for review.

ANSWER: Indeed, but the way the manuscript was presented to the reviewers, was not the way we designed it originally. In fact, it was actually just a Word-doc without the heading of Psych, as during submission process it was stated that it was not necessary to submit it a priori in Psych-heading. Our apologies either way, because it was clearly better for us to have designed the article completely in Psych-layout to begin with.

Reviewer 3 Report

Congratulations to the authors on choosing the topic, as well as the population they choose for the study. 

The Introduction is concisely written and clearly leads to the issue of the research.

The Methodology is well explained. 

The Results are presented well, and Discussion  follows the Results and well describes the limitations of the study. 

Author Response

REVIEWER 3:

Congratulations to the authors on choosing the topic, as well as the population they choose for the study. 

The Introduction is concisely written and clearly leads to the issue of the research.

The Methodology is well explained. 

The Results are presented well, and Discussion  follows the Results and well describes the limitations of the study. 

ANSWER: We thank the reviewer for their kind words and enthusiasm for the study.

Round 2

Reviewer 1 Report

All comments of the reviewer were corrected. The discussion, although it does not give a final answer to the question asked for the purpose of work, is carried out correctly and comprehensively.

It is advisable to describe in more detail the conclusions that result from the multi-threaded discussion

Author Response

We thank the author for their time in reviewing the article so thoroughly and the suggestions that have improved the article. 

"It is advisable to describe in more detail the conclusions that result from the multi-threaded discussion". We added: 

"To end this part of the discussion, we might conclude that in interpreting PTSS-results we must always look at of course, the methodology (e.g. How is PTSS measured? Who is included?), but equally important, the context in which the PTSS-results are found. In this study, for example, multiple contextual factors might have played a role, such as the geographical spread, the internal management of the organisation, the national context…"

Furthermore, we also added to the conclusion "An important result in this study was that while more funeral directors feared infecting their family and/or friends with COVID-19, it was the association between being infected themselves and development of PTSS that was significant. "